# Synergistic Push, Grasp and Active Vision with Evidential Learning for Manipulation-Enhanced Mapping in Confined Environments

Anonymous

## I. INTRODUCTION

We propose Multi-Skill Manipulation-Enhanced Mapping (MS-MEM), a hierarchical evidential framework for uncertainty-aware occlusion mapping that jointly reasons over active viewpoint selection, non-prehensile pushing, and prehensile grasping. MS-MEM combines a scene-level metric-semantic belief map with a local grasp representation based on a full-evidential extension of vMF-Contact (FE-vMF), which models both grasp affordance and directional uncertainty. Within a POMDP formulation, the framework predicts the outcomes of candidate actions and evaluates them using Disturbance- and Occlusion-aware Information Gain (DOIG), a unified objective that balances expected visibility improvement against scene disturbance across heterogeneous action skills. In this way, MS-MEM enables localized and controllable occluder removal, improving visibility while minimizing unnecessary global disturbance to the scene.

## II. PRELIMINARY: MANIPULATION-ENHANCED MAPPING

In general, Manipulation-enhanced Mapping (MEM) [1], [2] constructs an instance-level map space belief of a confined and cluttered shelf via active perception and strategic pushing.

Following the formulations from [1], at time $t$, let the current underlying environment be described by the state belief $\Phi_t$, which evolves under an executed action $a_t$ according to the scene dynamics. In a confined shelf with size $H \times W \times D$, the map belief estimates are given by $\Phi_t = \{\Phi_t^O, \Phi_t^S\}$, where $\Phi_t^O \in \mathbb{R}^{H \times W \times D}$ denotes the 3D occupancy belief map and $\Phi_t^S \in \mathbb{R}^{W \times D \times N_{cls}}$ denotes the top-down 2D semantic belief map, with $N_{cls}$ the number of semantic classes. The action $a_t$ belongs to one of the proposed push actions $\boldsymbol{p}_t$ or active view selection $v_t \in \mathcal{V}$ from a pool of viewpoint candidates $\mathcal{V}$.

To support calibrated belief propagation, MEM represents the probabilistic belief $\Phi_t$ in evidential form as $\lambda_t = \{\lambda_t^O, \lambda_t^S\}$. Here, $\lambda_t^O \in \mathbb{R}^{H \times W \times D \times 2}$ stores the occupancy evidence in the form of Beta distributions for each voxel. Given a voxel indexed by $u$ and its corresponding Beta parameters $\lambda_t^O[u] = [\alpha_{t,u}^O, \beta_{t,u}^O]$, the occupancy belief is modeled as

$$\Phi_t^O[u] \sim \text{Beta}(\alpha_{t,u}^O, \beta_{t,u}^O), \ \mathbb{E}[\Phi_t^O[u]] = \frac{\alpha_{t,u}^O}{\alpha_{t,u}^O + \beta_{t,u}^O}. \quad (1)$$

Similarly, $\lambda_t^S \in \mathbb{R}^{H \times W \times N_{cls}}$ stores the semantic belief in Dirichlet distributions. Let $\lambda_t^S[u] = [\eta_{t,u,c}^S]_{c=1}^{N_{cls}}$ denote the local Dirichlet parameters, the semantic belief is modeled as

$$\Phi_t^S[u] \sim \text{Dir}([\eta_{t,u,c}^S]_{c=1}^{N_{cls}}), \ \mathbb{E}[\Phi_{t,c'}^S[u]] = \frac{\eta_{t,u,c'}^S}{\sum_{c=1}^{N_{cls}} \eta_{t,u,c}^S}. \quad (2)$$

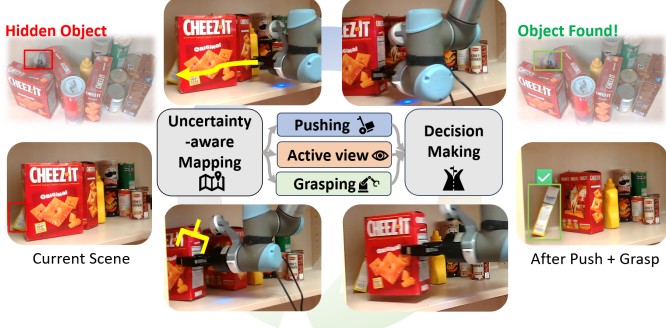

Fig. 1. Overview of MS-MEM.

Then, $\rho_{t,u} = \arg\max \mathbb{E}[\Phi_{t,c}^S[u]]_{c=1}^{N_{cls}}$ is the estimated class.

The belief propagation over time is instantiated through two Neural-Accelerated Belief Updates (CNABU)s, including: the observation CNABU $\sigma_o$, which updates the evidential belief $\lambda_t$ after receiving a new observation $o_t$; The push CNABU $\sigma_p$ predicts the post-push belief after $\boldsymbol{p}_t$. The evidential state is then propagated as

$$\lambda_{t+1} \leftarrow \sigma_o(\lambda_t, o_t, a_t), \quad \lambda_{t+1} \leftarrow \sigma_p(\lambda_t, a_t), \quad (3)$$

Overall, this enables calibrated reasoning over occlusions and manipulation-induced scene changes.

## III. METHODOLOGY

### A. System Overview

Our MS-MEM extends the standard MEM framework [1], [2] by strategically integrating prehensile grasping, non-prehensile pushing, and active view selection. As illustrated in Fig. 2, MS-MEM is organized as a closed-loop hierarchical framework, in which all subsequent modules operate on a shared evidential belief of the scene. At each time step, the framework considers three action branches conditioned on the current evidential scene belief $\Phi_t$: Viewpoint selection with Disturbance- and Occlusion-aware Information Gain (DOIG), Uncertainty-informed Push Selection (**UPS**), and Uncertainty-informed Grasp Selection (**UGS**).

Initially, all action candidates are compared within their individual modules by evaluating their post-action beliefs and the DOIG objective (Sec. III-B) to ensure fair comparisons across different action skills. Within the **UGS** module (Sec. III-C, illustrated in Fig. 3), the belief $\Phi_t$ is used to infer uncertainty-aware grasp hypotheses with the proposed Full-Evidential vMF-Contact (*FE-vMF*, Sec. III-C1). These hypotheses are further accumulated over time by the novel Full Evidential Uncertainty-guided Multi-view Grasp Fusion (*FE-UMGF*, Sec. III-C2). The **UGS** module evaluates these

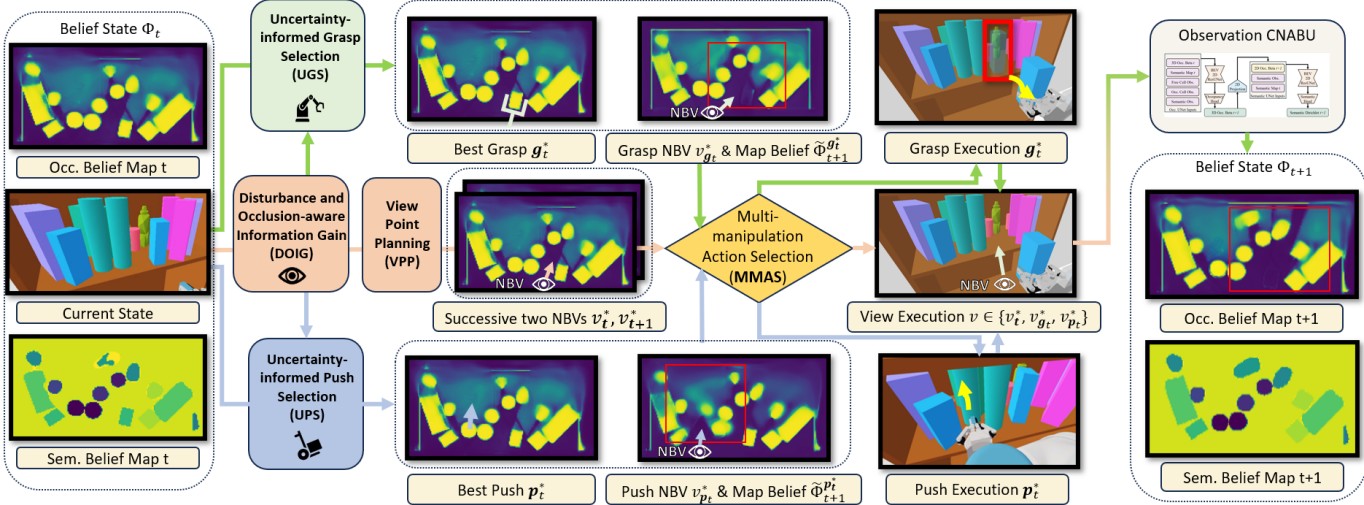

Fig. 2. Overall pipeline of MS-MEM.

uncertainty-aware grasp candidates by their respective DOIGs, while **UPS** evaluates push candidates with the same regime as [2] (Sec. III-D). Moreover, viewpoint actions are assessed via View Point Planning (**VPP** [1]), allowing their direct comparison to manipulation actions by Multi-Manipulation Action Selection (**MMAS**, Sec. III-E). Finally, after the execution of the chosen actions, the observation CNABU $\sigma_o$ updates the evidential map with the newly acquired observation $o_{t+1}$.

### B. Disturbance- and Occlusion-aware Information Gain (DOIG)

As an extension from volumetric information gain (IG) [3], DOIG serves as the common utility for all three action branches. Before its calculation, for manipulation actions $a_t \in \{p_t, g_t\}$, the system first predicts the post-action belief $\tilde{\Phi}_{t+1}^{a_t}$, and evaluates the volumetric IG of a subsequent NBV: $v_{t+1} \in \mathcal{V}$; For a pure view-change action $v_t$, the utility is evaluated as the sum of the immediate IG under the current belief $\Phi_t$ and the best subsequent IG after incorporating that observation $o_{t+1}$ (referred to as View Point Planning (VPP) [1]). In this way, all branches are evaluated over an equivalent two-step horizon, enabling a fair comparison.

*a) Occlusion-aware Information Gain (OIG):* Formally, the OIG [1] is defined as [1]

$$\text{OIG}(a_t) := \begin{cases} \zeta_o \Delta\text{H}(\Phi_t, \tilde{\Phi}_{t+1}^{a_t}) + \max_{v_{t+1} \in \mathcal{V}} \text{IG}(v_{t+1}|\tilde{\Phi}_{t+1}^{a_t}), a_t \notin \mathcal{V}, \\ \text{IG}(a_t|\Phi_t) + \max_{v_{t+1} \in \mathcal{V}} \text{IG}(v_{t+1}|\tilde{\Phi}_{t+1}^{a_t}), \quad a_t \in \mathcal{V}. \end{cases}$$

(4)

While manipulation actions inherently alter the map state, whose introduced uncertainty is captured by $\Delta\text{H}(\Phi_t, \tilde{\Phi}_{t+1}^{a_t})$, the resulting scene change to mapped and certain regions shall be additionally penalized for disturbing the scene stability. To achieve this, MS-MEM introduced Collateral Disturbance Constraint (CDC) into the OIG objective, giving the Disturbance and Occlusion-aware Information Gain (DOIG).

[1]For notational simplicity, OIG and DOIG both take $\tilde{\Phi}_{t+1}^{a_t}$ as input, while we assume they're automatically derived given $a_t$ depend on its action branch.

Concretely, CDC introduces a localized scene change penalty evaluated on the predicted post-action belief $\tilde{\Phi}_{t+1}^{a_t}$. First, we identify certain voxels of the scene using the semantic Dirichlet parameters $\lambda_t^S$. We calculate the semantic uncertainty over the evidential map and gather the set of semantically changed voxels $\mathcal{U}_{\text{diff}}$, as those (i) where the semantic uncertainty is below a threshold $\tau_u$ and (ii) the expected semantic class $\rho_{t,u}$ is changed:

$$\mathcal{U}_{\text{diff}}(\Phi_t, \tilde{\Phi}_{t+1}^{a_t}) := \left\{ u \; \middle| \; \frac{N_{cls}}{\sum_{c=1}^{N_{cls}} \eta_{t+1,u,c}^S} < \tau_u, \rho_{t+1,u} \neq \rho_{t,u} \right\}$$

(5)

To penalize destructive actions, we formalize the CDC via:

$$\max_{a_t} \text{OIG}(a_t) \quad \text{s.t.} \quad |\mathcal{U}_{\text{diff}}(\Phi_t, \tilde{\Phi}_{t+1}^{a_t})| \leq \Omega_c \quad (6)$$

With $\Omega_c$ as an allowable threshold of scene disturbance. The optimization is solved via the Lagrangian relaxation given the multiplier $\zeta_{\text{CDC}}$, so the constant $\Omega_c$ can be ignored [1]:

$$\text{DOIG}(a_t) := \text{OIG}(a_t) - \zeta_{\text{CDC}}|\mathcal{U}_{\text{diff}}(\Phi_t, \tilde{\Phi}_{t+1}^{a_t})|, \quad (7)$$

Notably, the DOIG reduces to OIG for pure view actions ($a_t \in \mathcal{V}$). This ensures the selected optimal action $a_t^*$ effectively balances targeted occluder reduction with the preservation of previously mapped evidential certainty.

### C. Uncertainty-informed Grasp Selection (UGS)

The system overview is depicted in Fig. 3 (A). In general, the **UGS** module first leverages a novel Full-Evidential vMF-Contact *FE-vMF* network which proposes evidential grasp hypotheses (Sec. III-C1). After uncertainty-aware temporal fusion with historical grasps through the *FE-UMGF* buffer (Sec. III-C2), it selects the best executable grasp from the candidates by computing their post-grasp beliefs $\tilde{\Phi}_{t+1}^{g_t}$ and comparing the corresponding DOIGs using Eq.(7).

*1) Full Evidential Grasp Learning:* As the core component of the **UGS** module, to enable uncertainty-aware grasp synthesis and selection, we propose the *Full-Evidential vMF-Contact (FE-vMF)* that quantifies uncertainty of the complete $SE(3)$

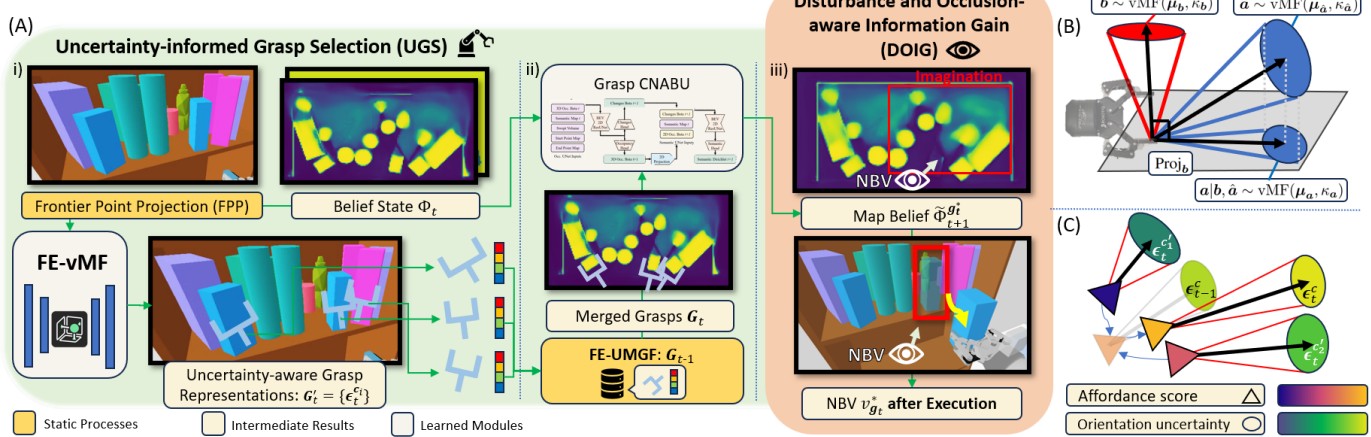

Fig. 3. (A) Uncertainty-informed grasp selection (**UGS**) pipeline; (B) Uncertainty-aware orthogonal projection for grasp orientations (Sec, III-C1); (C) Uncertainty-aware grasp fusion in FE-UMGF (Sec. III-C2)

grasp configuration. Unlike prior work [4], which merely restricts probabilistic modeling to the baseline element $p(\boldsymbol{b}|q, \boldsymbol{c})$, we treats them as fully distributional parametrizations in [4].

*a) Dual Orientational Uncertainty Representation:* As depicted in Fig. 3 (B), We model rotational uncertainty with two decoupled evidential parameterizations for the baseline $\boldsymbol{b}$ and the approach $\hat{\boldsymbol{a}}$ separately:

$$\hat{\boldsymbol{a}} \sim \text{vMF}(\boldsymbol{\mu}_{\hat{\boldsymbol{a}}}, \kappa_{\hat{\boldsymbol{a}}}), \quad \boldsymbol{b} \sim \text{vMF}(\boldsymbol{\mu}_{\boldsymbol{b}}, \kappa_{\boldsymbol{b}}). \quad (8)$$

Here, $\hat{\boldsymbol{a}}$ is an intermediate estimate used to construct the final approach distribution $\boldsymbol{a}|\boldsymbol{b}, \hat{\boldsymbol{a}} \sim \text{vMF}(\boldsymbol{\mu}_{\boldsymbol{a}}, \kappa_{\boldsymbol{a}})$, while enforcing the orthogonality constraint $\boldsymbol{a} \perp \boldsymbol{b}$. By projecting the distribution of $\hat{\boldsymbol{a}}$ onto the subspace orthogonal to $\boldsymbol{b}$, we get

$$\boldsymbol{\mu}_{\boldsymbol{a}} = \frac{\text{Proj}_{\boldsymbol{b}} \cdot \boldsymbol{\mu}_{\hat{\boldsymbol{a}}}}{\|\text{Proj}_{\boldsymbol{b}} \cdot \boldsymbol{\mu}_{\hat{\boldsymbol{a}}}\|}, \quad \kappa_{\boldsymbol{a}} = \kappa_{\hat{\boldsymbol{a}}} \|\text{Proj}_{\boldsymbol{b}} \cdot \boldsymbol{\mu}_{\hat{\boldsymbol{a}}}\|, \quad (9)$$

with $\text{Proj}_{\boldsymbol{b}} = \boldsymbol{I} - \boldsymbol{\mu}_{\boldsymbol{b}} \boldsymbol{\mu}_{\boldsymbol{b}}^{\top}$ denotes the orthogonal projection matrix onto the subspace perpendicular to $\boldsymbol{\mu}_{\boldsymbol{b}}$.

Predicting $\{\boldsymbol{\mu}_{\hat{\boldsymbol{a}}}, \kappa_{\hat{\boldsymbol{a}}}, \boldsymbol{\mu}_b, \kappa_b\}$ enables the model to express anisotropic uncertainty in the grasp orientation. For instance, two closely neighboring objects in a shelf may give rise to competing grasp hypotheses that share a similar approach direction but differ in their baseline directions.

*b) Evidential Affordance Representation:* Similar as evidential per-voxel occupancy parametrization as in Eq. (1), we treat the affordance probability $q$ as a random variable following Beta distributions: $q^{\boldsymbol{c}} \sim \text{Beta}(\alpha^{\boldsymbol{c}}, \beta^{\boldsymbol{c}})$. In practice, *FE-vMF* predicts evidence parameters $\boldsymbol{e}^{\boldsymbol{c}} = [e_\alpha^{\boldsymbol{c}}, e_\beta^{\boldsymbol{c}}]^T$ via a Softplus activation on the predicted raw logits $\hat{\boldsymbol{e}}^{\boldsymbol{c}} \in \mathbb{R}^2$:

$$\alpha^{\boldsymbol{c}} = e_\alpha^{\boldsymbol{c}} + 1, \quad \beta^{\boldsymbol{c}} = e_\beta^{\boldsymbol{c}} + 1, \quad \boldsymbol{e}^{\boldsymbol{c}} = \text{Softplus}(\hat{\boldsymbol{e}}^{\boldsymbol{c}}) \quad (10)$$

The total evidence quantifies the model's confidence. The predictive mean $\mathbb{E}[q^{\boldsymbol{c}}]$ and the epistemic uncertainty $u^{\boldsymbol{c}}$ are derived as:

$$\mathbb{E}[q^{\boldsymbol{c}}] = \frac{\alpha^{\boldsymbol{c}}}{S}, \quad u^{\boldsymbol{c}} = \frac{2}{S}, \quad S = \alpha^{\boldsymbol{c}} + \beta^{\boldsymbol{c}} \quad (11)$$

*c) Learning FE-vMF:* In each iteration, Given the current belief occupancy state $\Phi_t^O$ estimated by the observation CNABU $\sigma_o$, whose per-pixel occupancy confidence is calculated from Eq. (1), we extract the front surface point cloud via

frontier point projection (FPP), which uses ray-casting on the expected occupancy $\mathbb{E}[\Phi_t^O]$ to isolate only the surface points:

$$\text{pcd} = \text{FPP}(\mathbb{E}[\Phi_t^O]) \in \mathbb{R}^{N_{\text{pcd}} \times 3} \quad (12)$$

with $N_{\text{pcd}} = 1e4$ as the pre-defined number of sub-sampled points. To better capture the full evidential uncertainty, we leverage Point Transformer v3 (PTv3) [5] as the evidential backbone, which provides a strong permutation-invariant architecture while preserving fine-grained local geometry:

$$\{\boldsymbol{f}^{\boldsymbol{c}_i}\}_{i=1}^{N_{\text{pcd}}} = \text{PTv3}(\text{pcd}), \quad \boldsymbol{\epsilon}^{\boldsymbol{c}_i} = \text{MLP}(\boldsymbol{f}^{\boldsymbol{c}_i}). \quad (13)$$

$$\boldsymbol{\epsilon}^{\boldsymbol{c}_i} := \{\boldsymbol{\mu}_{\hat{\boldsymbol{a}}}^{\boldsymbol{c}_i}, \kappa_{\hat{\boldsymbol{a}}}^{\boldsymbol{c}_i}, \boldsymbol{\mu}_{\boldsymbol{b}}^{\boldsymbol{c}_i}, \kappa_{\boldsymbol{b}}^{\boldsymbol{c}_i}, \hat{e}^{\boldsymbol{c}_i}\} \quad (14)$$

$\boldsymbol{\epsilon}^{\boldsymbol{c}_i}$ denotes the evidential outcome on the potential contact point $\boldsymbol{c}_i$, decoded from per-point feature $\boldsymbol{f}^{\boldsymbol{c}_i}$ via a Multi-Layer Perceptron (MLP). Training details is in Sec VI-B

*2) Full Evidential Uncertainty-guided Multi-view Grasp Fusion (*FE-UMGF*):* As part of the **UGS** module, to enable seamless integration of *FE-vMF* into the POMDP-based MEM framework, which is built on *temporal state fusion* over states $\Phi_0, \ldots, \Phi_t$, we construct *temporal grasp fusion* by introducing Full Evidential Uncertainty-guided Multi-view Grasp Fusion (*FE-UMGF*). Inspired by [6], this module aims to integrate grasp predictions over time while maintaining consistency through systematic Bayesian updates.

Let $\boldsymbol{G}_{t-1}$ denote the global grasp buffer accumulated up to $t-1$. The current inference results $\boldsymbol{G}_t' = \{\boldsymbol{\epsilon}^{\boldsymbol{c}_i'}\}_{i=1}^{N_{\text{pcd}}}$ from Eq. (14) represent the newly inferred grasps from the occupancy belief $\lambda_t^O$. The overall temporal update is denoted as: $\boldsymbol{G}_t \leftarrow \textit{FE-UMGF}(\boldsymbol{G}_{t-1}, \boldsymbol{G}_t')$.

Deviating from [6], *FE-UMGF* aims to generalize the fusion process to account for the full evidential uncertainty. For an old fused grasp $\boldsymbol{\epsilon}_{t-1}^{\boldsymbol{c}} \in \boldsymbol{G}_{t-1}$ located at the cluster center $\boldsymbol{c}$, its accumulated Beta evidence parameters $\alpha_t^{\boldsymbol{c}}$ and $\beta_t^{\boldsymbol{c}}$ are updated by aggregating the evidence from all nearby candidates $\boldsymbol{c}'$ assigned to its neighbouring cluster $C_t$ [2]:

$$\alpha_t^{\boldsymbol{c}} = \gamma \alpha_{t-1}^{\boldsymbol{c}} + \sum_{\boldsymbol{c}' \in C_t} \alpha_t^{\boldsymbol{c}'}, \quad \beta_t^{\boldsymbol{c}} = \gamma \beta_{t-1}^{\boldsymbol{c}} + \sum_{\boldsymbol{c}' \in C_t} \beta_t^{\boldsymbol{c}'} \quad (15)$$

[2]Due to space constraints, we refer interested readers to [6], Eq. (8)-(12), for a detailed exposition of the grasp clustering and fusion procedure.

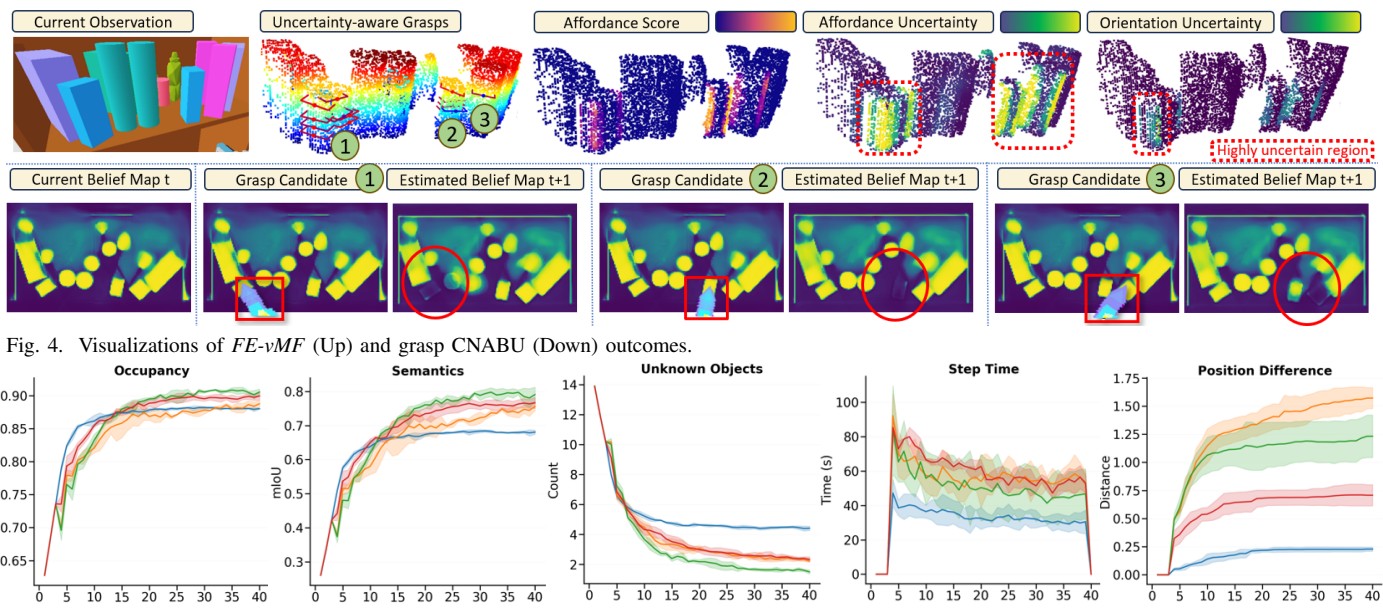

Fig. 4. Visualizations of *FE-vMF* (Up) and grasp CNABU (Down) outcomes.

Fig. 5. Simulation experiments compared to: (i) Pure grasping; (ii) Pure pushing (MEM [1]); (iii) Our approach w/o CDC penalty, which uses vanilla OIG objective instead of DOIG); (iv) Our complete approach.

TABLE I
PERFORMANCE AT THE LAST STEP (= 40).

| Method | Occupancy ↓ | Semantics ↑ | Position Difference ↓ |
|---|---|---|---|
| Grasp only | $0.88 \pm 0.001$ | $0.68 \pm 0.006$ | $0.23 \pm 0.018$ |
| Push only [1] | $0.89 \pm 0.005$ | $0.76 \pm 0.011$ | $1.57 \pm 0.094$ |
| w/o CDC | $0.91 \pm 0.005$ | $0.79 \pm 0.022$ | $1.23 \pm 0.189$ |
| Ours | $0.90 \pm 0.006$ | $0.77 \pm 0.016$ | $0.71 \pm 0.094$ |

Then, the $\mathbb{E}[q_t^c]$ can be calculated by Eq. (11). By decaying historical evidence with decay $\gamma$, we implicitly assign higher relative weights to novel belief states $\Phi_t$ and retain the grasp outcomes from previous states $\{\Phi_{t-1}, \Phi_{t-2}, ...\}$.

For the orientational elements, we build upon the clustering and fusion procedure from [6], while independently aggregate both $\hat{a}$ and $b$ within each local cluster $C_t$. The final approach vector $a$ is recovered via the projection defined in Eq. (9).

*3) Final Grasp Selection:* Finally, the optimal grasp $g_t^*$ and associated future viewpoint $v_{g_t}^*$ is selected to maximize the DOIG on the predicted post-grasp belief $\tilde{\Phi}_{t+1}^{g_t^c}$, by maximizing Eq. (7). In this way, **UGS** selects the best grasp and the subsequent NBV based on their confidence, executability, and expected contribution to reduce the map uncertainty.

### D. Uncertainty-informed Push Selection (**UPS**)

Similar to **UGS**, the **UPS** module generates targeted non-prehensile pushing actions $p_t$ to uncover occluded regions and their corresponding NBVs $v_{p_t}^*$ using uncertainty-aware visibility corridors [2].

### E. Multi-Manipulation Action Selection (**MMAS**)

As the final stage, **MMAS** module serves as the central active decision-making mechanism that selects the optimal action $a_t^*$ by evaluating DOIG (Eq. (7)) of all skills.

## IV. EXPERIMENTS

In the experiments, we compare MS-MEM with single-skill baselines on mapping performance and scene disturbance, and visualize *FE-vMF* and grasp CNABU outputs for uncertainty-aware grasp prediction and post-grasp belief updates. The training details can be found under Sec. VI-C.

### A. Simulation Experiments

*a) Mapping Performance:* Fig. 5 presents the results of our simulation experiments. We compare our method against: (i) *Grasp Only*: pure grasping; (ii) *Push Only*: pure pushing that corresponds to the vanilla MEM [1], which was proven to outperform pure active view planning; (iii) *w/o CDC*: our method without the CDC penalty, where the standard OIG is used instead of DOIG; and (iv) *Ours*: our full approach. **VPP** branch is activated for all baselines.

In general, *Ours* achieves high mapping accuracy (90% occupancy IoU and 77% semantic IoU) and low scene disturbance (0.71) at the same time.

*b) Evidential Grasp Learning Outcomes:* Fig. 4 shows the visualization of *FE-vMF* (up) and grasp CNABU (down) outputs. The high affordance uncertainty appears mainly on partially visible objects, while the orientation tends to geometric ambiguity. The lower rows show that grasp CNABU predicts candidate-specific post-grasp belief updates. Starting from the same current belief map $\Phi_t$, each grasp candidate leads to individual estimated belief maps $\tilde{\Phi}_t^g$.

## V. CONCLUSION

In this paper, we present MS-MEM, an evidential framework for uncertainty-aware mapping via active decision-making under grasp, push and active view selection. By extending MEM with the proposed *FE-vMF* for evidential grasp representation learning, *FE-UMGF* grasp fusion, and the unified DOIG objective, MS-MEM enables direct comparison of heterogeneous action skills under a shared evidential belief representation. Our results show that jointly leveraging pushing and grasping provides clear advantages over single-skill baselines for evidential metric-semantic mapping.

## VI. APPENDIX

### A. Hierarchical Bayesian Grasp Representation

We formulate the 6-DoF grasp synthesis as a hierarchical Bayesian inference following [4]. For a given contact point $c \in \mathbb{R}^3$, a grasp candidate $g = \{q, a, b, w\}$ is decomposed into: (i) affordance (or grasp quality in [7]) $q \in [0, 1]$; (ii) a primary baseline vector $b$ and a secondary approach vector $a$, so that $a, b \in S^2$ and $a \perp b$; (iii) the grasp width $w \in R$.

The joint probability density of the grasp configuration is factorized according to the hierarchical contact grasp representation following [8]:

$$p(g|c) = \int \underbrace{p(q|c)}_{\text{Affordance}} \cdot \underbrace{p(b|q, c)}_{\text{Baseline}} \cdot \underbrace{p(a|b, q, c)}_{\text{Approach}} \, dq \quad (16)$$

In this hierarchy, conditioning on a viable affordance, the baseline probability $p(b|q, c)$ defines the directional density function via a von Mises-Fisher (vMF) distribution [4]:

$$p(b|\mu, \kappa) = \text{vMF}(\mu, \kappa) := \mathcal{C}(\kappa) \exp(\kappa \mu^T b), \quad (17)$$

where $\mu \in S^2$ is the mean direction and $\kappa > 0$ is the concentration parameter, normalized by $\mathcal{C}(\kappa) = \frac{\kappa}{4\pi \sinh \kappa}$.

### B. Objective for FE-vMF

To train the *FE-vMF*, we minimize a composite objective:

$$\mathcal{L}_{FE}^{c_i} = q^\dagger \left( \mathcal{L}_{a^\dagger}^{\text{BL}}(\mu_{\hat{a}}^{c_i}, \kappa_{\hat{a}}^{c_i}) + \mathcal{L}_{b^\dagger}^{\text{BL}}(\mu_b^{c_i}, \kappa_b^{c_i}) \right) + \mathcal{L}_{q^\dagger}^{\text{EDL}}(\hat{e}^{c_i}) \quad (18)$$

with one-hot ground truths grasp score $q^\dagger = \mathbb{I}(c_i \approx c^\dagger)$ when the point $c^\dagger$ is within 2mm to $c_i$. Here, we minimize the Bayesian losses $\mathcal{L}_{(\cdot)}^{\text{BL}}$ towards the ground-truth vectors $a^\dagger, b^\dagger$ respectively following [4]. Training the affordance minimizes the Beta evidential loss $\mathcal{L}_{q^\dagger}^{\text{EDL}}$ towards $q^\dagger$ following [9].

### C. Training details

*a) Data Generation:* The data generation pipeline is powered by the Pybullet simulation engine [10], where the setup contains a confined shelf space with a UR5 robotic manipulator, equipped with a Robotiq 2F-85 gripper. The simulation and real-world setups (Fig. 6) are identical except for the perception system. In simulation, a set of projection-based pinhole cameras is used to approximate the wrist-mounted RealSense L515 camera. The predefined viewpoint candidates with a total number of $|\mathcal{V}| = 300$ are kept consistent across both environments.

*b) Model Training Details:* For training *FE-vMF*, we generated $4 \times 10^3$ simulated shelf scenes with randomly arranged objects with occupancy fractions ranging from $30 - 45\%$. To collect collision-free ground truths grasps $\{g^\dagger\}$ in these scenarios, we: (i) generate $1e5$ 6-DoF antipodal grasps for each object in isolation, and (ii) filter them in the scenes by removing grasps that collide with the shelf or surrounding objects. At this end, each sample contains a merged point cloud from 3 to 10 random viewpoints and $100 - 400$ ground truth grasps. The training of the objective in Eq. (18) uses a single RTX4090 GPU with batch size 16, optimized by AdamW [11] with learning rate $1e - 4$ and cosine annealing.

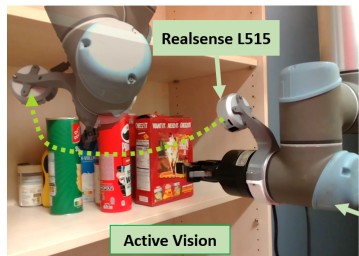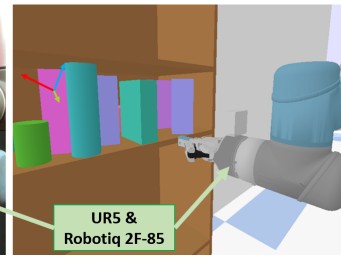

Fig. 6. Simulation and real-world setups.

The grasp CNABU $\sigma_g$ training is designed similarly to the training in [2]. With the observation CNABU network as a starting point, a network for grasp prediction has been trained. The training dataset consisted of 7000 simulated grasps, with each entry consisting of partial maps before the grasp, ground truth maps before and after the grasp, and a swept volume representing the gripper movement during the grasp. Besides the removal of motion parametrization, the training framework is the same as which has been used in [2], for the push CNABU. The training utilized on a single RTXA6000. The lowest validation losses were achieved after around 20 episodes, with a learning rate of $5e - 4$.

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
