# OpenReview forum: "Synergistic Push, Grasp and Active Vision with Evidential Learning for Manipulation-Enhanced Mapping in Confined Environments"
_IEEE.org/ICRA/2026/Workshop/Manipulation_Robustness — ICRA 2026_

### Official Review · Reviewer_Y7QE · 2026-05-03
**Uncertainty-aware manipulation-enhanced mapping, but real-world robustness evidence is limited**

**Rating:** 6
**Confidence:** 4

**Review:**

This paper presents MS-MEM, a manipulation-enhanced mapping framework that selects among active viewpoint changes, pushing, and grasping in confined cluttered environments. The problem is relevant to the workshop, and the main idea of comparing these heterogeneous actions under a shared disturbance- and occlusion-aware information gain objective is interesting. I particularly like the motivation for penalizing scene disturbance, since maximizing information gain alone may encourage aggressive manipulation that unnecessarily changes an already partially mapped scene. The uncertainty-aware grasp representation and multi-view grasp fusion are also reasonable additions for partially observed cluttered scenes.

The main weakness is that the experimental evidence is still limited relative to the robustness motivation. Most quantitative results appear to be simulation-based, while the real-world setup is only briefly shown. I would like to see more analysis under realistic perception and manipulation failures, such as missing depth, reflective or dark objects, inaccurate push outcomes, grasp failures, and out-of-distribution object configurations. This concern is also relevant to the use of a wrist-mounted RealSense L515, whose depth quality may be sensitive to material properties and close-range operation in confined manipulation settings. The reported results support a useful mapping–disturbance trade-off, but the paper should state this more carefully, since the variant without the disturbance penalty achieves slightly higher occupancy and semantic IoU. Overall, I would support accepting the paper for workshop discussion, but I would not view the current evidence as sufficient to support strong claims about real-world robustness.

---

### Decision · Program_Chairs · 2026-05-21

Accept